# Academic enjoyment, behavioral engagement, self-concept, organizational strategy and achievement in EFL setting: A multiple mediation analysis

Xia Kang[1,2], Yajun Wu [3]*

1 School of Mathematics, Yunnan Normal University, Kunming, China, 2 Teacher Education and Learning Leadership Unit, Faculty of Education, The University of Hong Kong, Hong Kong, 3 School of Foreign Languages and Literature, Yunnan Normal University, Kunming, China

* wuyajun1225@163.com

**Data Availability Statement:** All relevant data are within the manuscript.

**Funding:** The authors received no specific funding for this work.

## Abstract

Motivated by the positive psychology movement in the English as Foreign Language (EFL), existing studies have demonstrated that subject-related enjoyment has a positive correlation with academic achievement. However, quite a few studieshave examined why academic enjoyment can predict positive academic achievement. This study aimed to investigate whether behavioral engagement, self-concept, and organizational strategy mediated relations between academic enjoyment and achievement in EFL setting. 528 Chinese secondary school students (Male: $N = 280$; Female: $N = 248$) participated in the survey and completed the questionnaires on EFL-related enjoyment, behavioral engagement, self-concept, organizational strategy, and academic performance. Structural equation model (SEM) analyses showed that students' academic enjoyment positively predicted their English achievement. Academic engagement, self-concept, and organizational strategy had parallel multiple mediating effects between academic enjoyment and English achievement. Multi-group SEM analysis demonstrated that the model had invariance across genders, indicating that the model is applicable to both male and female students. Limitations and implications are discussed.

## Introduction

This study aimed to further understand the association between achievement emotions and academic achievement in the context of English as a Foreign Language. Previous research has indicated that achievement emotions are not only a by-product of the learning process and/or achievement, but also play a critical role in subsequent academic achievement via mediators such as motivation, academic engagement, learning strategies, and competence beliefs [e.g., 1–4]. According to the control-value theory (CVT), positive emotions such as enjoyment, hope, and pride would benefit the improvement of academic achievement, while negative emotions such as anxiety, boredom, and hopelessness have negative impacts on student's academic

**Competing interests:** The authors have declared that no competing interests exist.

achievement [5–7]. Inspired by positive psychology, more and more studies were conducted to explore the positive effect of positive achievement emotions on academic achievement in the field of education. Specifically, the influence of academic enjoyment, as the most frequent and intense positive emotion [8, 9], on academic success and well-being has been widely explored in STEM fields (science, technology, engineering, and mathematics) [10, 11]. Recently, in addition to some studies focusing on language anxiety, interest in discussing positive achievement emotions is also growing (e.g., academic enjoyment)in the EFL setting [12–14]. Furthermore, by quizzing undergraduates on their achievement emotions about mathematics, Latin, German, and English, [15] found that students' achievement emotions are domain-specific. More research on the correlation between academic enjoyment and subsequent achievement is therefore needed in the EFL context. Besides, studies that examine the relationship between academic enjoyment and achievement have took university students or senior high school students as participants [11, 14, 16] and less attention has been paid to the population of secondary school students. To address these limitations, the present study aimed to examine the association between academic enjoyment and achievement as well as the mediaton mechanism between these two constructs in EFL context, in a sample of secondary school students.

## Literature review

### Academic enjoyment and English achievement

Control-value theory of achievement emotions postulates that control and value appraisals are the proximal antecedents of achievement emotions, which, in turn, affect students' academic achievement [5, 17]. Based on control-value theory, researchers documented that achievement emotions would affect academic performance via mediators such as cognitive resources, achievement goals, motivational engagement, and strategy use [e.g., 18, 19]. In addition, the control-value theory [e.g., 20] and empirical evidence [e.g., 4, 21] demonstrated the reciprocal relationships between achievement emotions and academic self-concept, which implies that academic self-concept would also be one possible mediator between achievement emotions and academic performance. Achievement emotions refer to emotions directly related to achievement activities or achievement outcomes. Precisely, previous research has identified nine discrete emotions (i.e., enjoyment, hope, pride, relief, anxiety, boredom, hopelessness, shame, and anger) in three contexts (i.e., classroom, homework, and examination) [15, 22]. Moreover, achievement emotions are domain-specific [15]. Therefore, researchers [e.g., 22, 23] suggested that both the subject and context should be taken into consideration when discussing a particular kind of achievement emotion. Generally, positive emotions, such as enjoyment, would play a critical role in the learning process because positive emotions are believed to be closely related to students' self-discipline, learning strategies, motivation, and the activation of cognitive resources [2].

Academic enjoyment is a positive predictor of achievement [16, 24]. However, little is known about the mediation mechanism between these two constructs. For example, [25] found that the higher level of students' EFL-related enjoyment, the higher their attitude towards EFL and their enthusiasm to engage in speaking practice. Moreover, by conducting a correlation analysis, [26] compared the predictive effects of enjoyment and anxiety on EFL achievement and reported that the effect of enjoyment on EFL achievement was higher than that of anxiety. Researchers have recognized the positive effect of academic enjoyment on EFL achievement [14, 16, 24, 25]. Few studies have been conducted to explore the mediation mechanism between academic enjoyment and achievement in an EFL context in China. Therefore, it is of significance to explore the mediators underpinning the linkage between academic

enjoyment and English achievement in the EFL context in a sample of Chinese secondary school students.

## The mediating roles of behavioral engagement, self-concept, and organizational strategy underpin the association between academic enjoyment and achievement

**Mediation via behavioral engagement.** Behavioral engagement is a multidimensional concept that refers specifically to student classroom conduct, participation in school-related activities, and interest in an academic task [27, 28]. Among them, students' classroom conduct and participation in school-related activities are the passive aspects of behavioral engagement for activities in these two settings were assigned by teachers, and students were practically driven by teachers' expectations. While interest in academic tasks is the active aspect of behavioral engagement because a student would energetically raise questions or participate in discussions in the classroom context.

Existing studies have shown the predictive effect of behavioral engagement on academic achievement [3, 29, 30]. For example, [29] explored the correlation between student engagement and academic achievement and noted that students with a high level of engagement perform better than those with a low level of engagement. Besides, research has also demonstrated the predictive effect of achievement emotions on students' engagement [31–33]. For example, [31] examed the promoting effect of positive achievement emotions (e.g., enjoyment) on students' academic engagement and revealed that the higher the level of students' positive emotions, the more engaged they would be in learning activities. Taken together, the present study proposed the structural mediational model that behavioral engagement plays a mediating role between academic enjoyment and achievement in an EFL context.

**Mediation by self-concept.** Self-concept refers to one's self-perceptions that are formed through experience within the environment [34]. In line with this definition, [35] defined academic self-concept as students' perceptions about themselves in achievement situations. More precisely, academic self-concept is about students' self-perceptions about their academic ability, that is, students' self-evaluation of whether they would achieve the specific achievement goals [36]. [37] reported that mathematics-related self-concept differed from the German- or general-school-related self-concept. This indicates that academic self-concept is domain-specific, and the measurement of academic self-concept should be conducted in a specific subject domain (e.g., mathematics, EFL, and science).

According to the CVT, positive self-concept implies students' control over the learning activities, which, in turn, predicts students would experience enjoyment in the related learning activities [5]. [2] examined the mediating effect of academic self-concept on mathematics-related achievement and enjoyment. Similarly, in a sample of nine graders' mathematics learning, [38] confirmed that academic self-concept was a complete mediator between academic achievement and enjoyment. Moreover, by conducting a longitudinal study relating to mathematics, [4] proved that academic self-concept fully mediated the link between enjoyment and achievement. In sum, the CVT model as well as empirical evidence suggested the reciprocal relationships between academic self-concept and achievement emotions [4, 20, 21], which means that academic-concept would be one possible mediator between academic enjoyment and academic performance. The mediating effect of academic self-concept between enjoyment and achievement was mainly verified in the context of mathematics, however, little work has been done to test this mediation model in the EFL context.

**Mediation through organizational strategy.** Organizational strategy, which refers to the selecting, sequencing, outlining, reordering, and summarizing important learning contents, is

a commonly used learning strategy for foreign language learners [39]. For example, [40] argued that among the various foreign language learning strategies, those are related to comprehension, organization, and elaboration are most commonly used by foreign language learners in the context of East Asian Confucious culture.

Previous studies have verified the associations between learning strategies and academic achievement and positive achievement emotions, such as academic enjoyment. For example, learning strategies have been found to positively predict L2 proficiency through the mediator of academic engagement [41]. Besides, taking students from grade 5 to grade 10 as participants, [42] explored the correlation between academic enjoyment and learning strategies and found that the higher the level of academic enjoyment, the more likely they were to adopt complex learning strategies. By exploring the correlation between achievement emotions and cognitive/meta-cognitive strategies, [43] found that organizational strategy was positively correlated with academic enjoyment. As academic enjoyment, learning strategies (e.g., organizational strategy), and academic achievement are all domain-specific [44–46], the present study examined the mediating effect of organizational strategy between academic enjoyment and achievement in an EFL context to expand the scope of this mediating model.

## Aims and hypotheses of the present study

The present study aimed to examine the mediating mechanisms between academic enjoyment and achievement in a sample of Chinese secondary school students aged 11 to 14 years old in the EFL context. As the constructs of academic enjoyment, academic engagement, self-concept, organizational strategy, and achievement are domain-specific [42, 44, 46, 47], they were measured in the English subject. In Mainland China, English is a compulsory subject in the whole course of education (from primary grade 3 to Ph. D. education) [48, 49], and it is of significance for both further study and promotion at work [50]. The Ministry of Education of the People's Republic of China stipulates that English should be introduced at primary three, however, to gain a competitive advantage, English was usually introduced as early as primary one or in the kindergarten phase in the high socioeconomic status schools [51]. Accordingly, gaining academic success in EFL, especially at the secondary school level, has been central to research efforts [11, 14, 16, 52]. However, researchon how positive achievement emotions (e.g., enjoyment) promote academic achievement in the EFL context needs further expanded, which may be helpful for understanding the mechanism of positive emotions on academic performance.

Necessarily, the present study filled the gap by exploring the mediation mechanism that underpins the association between academic enjoyment and English achievement among Chinese secondary school students in the EFL context. Four hypotheses were examined:

*Hypothesis 1*: Academic enjoyment has a positive predictive effect on Chinese secondary school students' English achievement.

*Hypothesis 2*: Behavioral engagement partially mediates the link between EFL-related enjoyment and achievement.

*Hypothesis 3*: Self-concept plays a partial mediating role in the relationship between EFL-related enjoyment and achievement.

*Hypothesis 4*: Organizational strategy partially mediates the association between EFL-related enjoyment and achievement.

## Methods

### Participants and procedure

**Ethics statement.**    The present study involving human participants were reviewed and approved by the Human Research Ethics Committee of the University of Hong Kong. Before

conducting the survey, participants and their English teachers provided their written informed consent to participate in this study. Only the data of participants who agreed to take part in the questionnaire survey would be used. Besides, verbal informed consent was also obtained from participants' parents or guardians. Students' participants in this questionnaire was entirely voluntary and they could stop and withdraw from this survey at any time.

**Participants.** A total of 528 seventh- and eighth-grade students (male n = 280, female n = 248, missing n = 22; mean age = 13.65, SD = 0.61) from one secondary school in Kunming City, a new first-tier city in the Chinese setting were selected as the participants. Participants were all from the main urban area of Kunming City. And as far as the socioeconomic status is concerned, participants were majorly from the middle class. In addition, given that the participant were mainly composed of Han students, the English learning environment was influenced by the Confucian-heritage culture [53–55].

**Design and procedure.** Students were recruited to complete the questionnaire during class with the help of a collaborator and English teachers and were told that they have the right to opt-out of the questionnaire if they do not wish to attend. Even if participants refuse to take part in the questionnaire, it will not affect them in any way. Expect for those who asked for leave, all the seventh- and eighth-graders in the school participated in the survey. The questionnaire took the participants about 20 minutes.

## Measures

**Academic enjoyment.** Participants' classroom-related enjoyment in EFL context was assessed by the five related items that were adapted from the Achievement Emotions Questionnaire [19] to reflect students' sense of satisfaction in the settings of their English classroom learning (e.g., 'I like English class'). Participants responded using a five-point scale (1 = strongly disagree to 5 = strongly agree). All factor loadings were significant ($p < 0.001$), and the Cronbach's alpha reliability for EFL-related enjoyment ranged from .56 to .86. The internal consistency reliability of the scale of academic enjoyment was .866. Confirmatory factor analysis showed that the model fit of this scale was good: $\chi^2/df = 3.107$, CFI = 0.992, TLI = 0.983, RMSEA = 0.063, and SRMR = 0.021. In the SEM analysis, EFL-related enjoyment was treated as a latent variable.

**Behavioral engagement.** *Engagement vs. Dissatisfaction with Learning Questionnaire* [56] was adapted to measure participants' behavioral engagement. The behavioral engagement was measured by four items and all items were adapted for English lessons or learning activities (e.g., "I try hard to do well in English class"). A 5-point Likert scale ranging from 1 (strongly disagree) to 5 (strongly agree) was used to describe participants' willingness to devote themselves to English learning. The higher the score, the more willing participants are to devote themselves to English learning. The internal consistency reliability of the behavioral engagement scale was .848. Item factor loadings ranged between .65 and .85 and were significant at $p < 0.001$. Besides, this scale showed a good fit: $\chi^2/df = 4.993$, CFI = 0.991, TLI = 0.973, RMSEA = 0.087, SRMR = 0.016. Behavioral engagement was also modeled as latent variable in SEM analysis.

**Academic self-concept.** The academic self-concept was measured with 5 items (e.g., "I am good at English") that adapted from the Program for International Student Assessment 2015 (PISA 2015) [57]. Participants responded on a 5-point Likert scale (1 = strongly disagree, 5 = strongly agree) such that a high score indicated that participants have strong self-perception of their English ability. Item factor loadings ranged from .60 to .87 and were significant at $p < 0.001$. The Cronbach's alpha for the academic self-concept was .866. In addition, the confirmatory factor analysis demonstrated that the scale of academic self-concept has an adequate model fit: $\chi^2/df = 1.971$, CFI = 0.996, TLI = 0.992, RMSEA = 0.043, SRMR = 0.014.

**Organizational strategy.** The organizational strategy was measured using five items from the Goal Orientation and Learning Strategies Survey (GOALS-S) [39]. Items were adapted to refer to English learning activities (e.g., "I study English by organizing my English notes"). Items of the organizational strategy were rated on a 5-point Likert scale ranging from 1 to 5 with higher scores indicating a stronger endorsement of organizational strategy in students' English learning activities. Item factor loadings ranged from .70 to .87 and were significant at p<0.001. The internal consistency reliability of organizational strategy was .897. Moreover, the model fit of the organizational strategy scale was good as $\chi^2/df$ = 3.642, *CFI* = 0.991, *TLI* = 0983, *RMSEA* = 0.071, *SRMR* = 0.016 This measure was posited as a latent variable in SEM analysis.

**English achievement.** Participants' English scores on the final course exam were used to represent their English achievement. The final course exam was uniformly developed and scored by the district education bureau according to the textbook and the syllabus. The total score of the test paper is 120 points. The higher the score, the higher the participants' English achievement. In the present study, English achievement was treated as an observed variable.

## Data analysis

Mplus 8.3 [58] was used to analyze the data in three stages. First, confirmatory factor analyses were conducted to verify the reliability and validity of each construct. Second, latent SEM was performed to test the path coefficients. Precisely, the mediating effects of behavioral engagement, self-concept, and organizational strategy were estimated by employing the bootstrap approach with bootstrapped confidence intervals of 95%. Third, multi-group SEM was applied to examine the measurement invariance of between-constructs across genders (i.e., male and female students).

To measure the measurement invariance across genders, three hierarchical processes proposed by [59] were performed, including configural, metric and scalar invariance. To test the configural invariance, factor loadings and thresholds across the contrasting groups were set free. Once configural invariance was established, then metric invariance across groups would be tested by equally constraining the factor loadings. And if metric invariance was proved, scalar invariance across groups would be estimated by equally constraining both factor loadings and intercepts of all items. Measurement invariance across groups would be established only if the following two conditions are met: (a) the overall model fit is acceptable [60], and the value of ΔCFI between two nested models is smaller than or equal to 0.01 [61].

## Results

### Common method bias

In this study, Harman's single-factor test was carried out to assess the common method bias [62]. The results of single-factor CFA were $\chi^2/df$ = 10.014, *CFI* = 0.773, *TLI* = 0.744, *RMSEA* = 0.131, *SRMR* = 0.074, indicating that the model fit was poor. This means that there is no significant common method bias in the dataset of the present study.

### Descriptive statistics and correlations

Table 1 shows the results of descriptive statistics and correlational analyses on the variables of academic enjoyment, behavioral engagement, self-concept, organizational strategy, English achievement, and gender. The results demonstrated that academic enjoyment, behavioral engagement, self-concept, and organizational strategy were positively correlated with English achievement, while the correlation between gender and academic enjoyment was not significant.

**Table 1. Descriptive statistics and zero-order bivariate correlational coefficients among the variables ($N$ = 528).**

| Variables | 1 | 2 | 3 | 4 | 5 | 6 |
|---|---|---|---|---|---|---|
| 1. Enjoyment | - | | | | | |
| 2. Behavioral engagement | .663** | - | | | | |
| 3. Self-concept | .617** | .627** | - | | | |
| 4. Organizational strategy | .588** | .671** | .582** | - | | |
| 5. Achievement | .338** | .424** | .573** | .291** | - | |
| 6. Gender | .002 | .021 | .023 | .073 | .111* | - |
| Mean | 3.752 | 2.957 | 2.587 | 2.874 | 90.691 | - |
| SD | .717 | .556 | .614 | .588 | 21.797 | - |
| Cronbach'α | .866 | .848 | .866 | .897 | - | - |

*Note.*

**p<0.01

*p<0.05.

## Examining the structural model

**Total effect modeling analysis.** Hypothesis 1 was verified by testing the linkage between academic enjoyment and English achievement without considering the intermediate variables-behavioral engagement, self-concept, and organizational strategy. Regression analysis showed that academic enjoyment had a positive predictive effect on English achievement, with the non-standardized regression coefficient being significantly non-zero ($\beta$ = 11.657, $p$<0.001). Furthermore, the standardized regression coefficient ($\beta$ = 0.371) also indicated that $H_1$ is valid.

**Mediation effect modeling analysis.** The proposed hypothetical model was tested by using the latent variable SEM, and the standardized path coefficients were presented in Fig 1. Results indicated that the proposed structural model had good fit indices, with $\chi^2$(164) = 549.917, $\chi^2/df$ = 3.353, $CFI$ = 0.939, $TLI$ = 0.929, $RMSEA$ = 0.067, $SRMR$ = 0.069.

The point estimates of standardized mediation effect of Path one (academic enjoyment→ behavioral engagement→ English achievement) were 0.214, and both the bias-corrected and accelerated 95% confidence interval (BCa 95% CI) were [0.078, 0.362], indicating that the mediating effect of behavioral engagement was significant. For the second Path (academic enjoyment→ self-concept→ English achievement), the point estimates of standardized

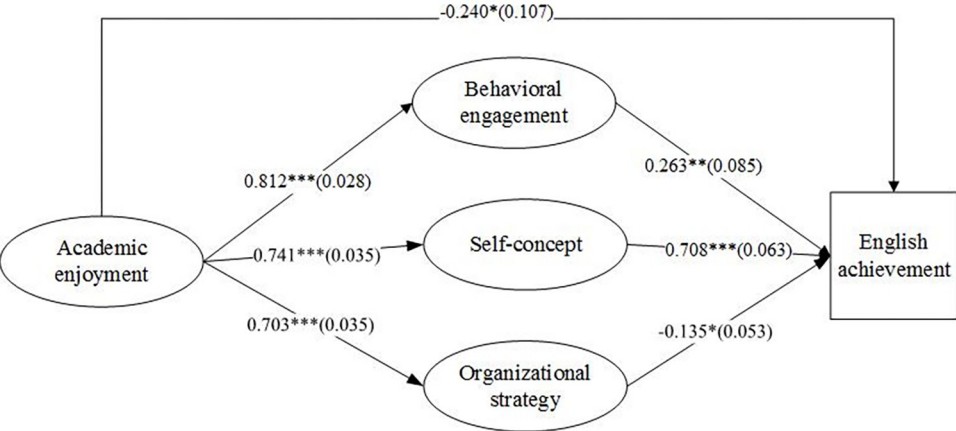

**Fig 1. Multiple mediation models between academic enjoyment and English achievement.** *Note.* Standardized path coefficients and standard error are presented. *p < 0.05, **p < 0.01, ***p < 0.001.

mediation effect = 0.525, and the BCa 95% CIs were [0.423, 0.657] and [0.419, 0.647] respectively, showing that self-concept was a significant mediator in the association between academic enjoyment and English achievement as zero was not contained. The point estimates of standardized mediation effect of Path three (academic enjoyment→ organizational strategy→ English achievement) = -0.094, and the BCa 95% CIs were [-0.173, -0.023] and [-0.172, -0.022] respectively. The BCa 95% CI of Path three did not contain zero, indicating that the mediation effect of the organizational strategy was significant.

Results of the pairwise multiple comparison analyses (see Table 2) indicated that the mediation effects of the three mediators were all significant, and the strengths of these three mediation effects were different. First, the unstandardized estimate for the strength comparison of behavioral engagement and self-concept was -10.137, and the BCa 95% CIs were [-16.096, -4.248] and [-16.103, -4.252], indicating that the mediation effect of self-concept was significantly different from (zero didn't occur in the BCa 95% CIs) and stronger than the mediation effect of behavioral engagement. Second, the unstandardized estimate for the comparison of the strength of the two mediation effects between behavioral engagement and organizational strategy was 10.026, and zero didn't occur between the BCa 95% CIs ([4.600,16.592] and [4.587, 16.587]), which suggests that the behavioral engagement's mediation effect is significantly stronger than that of organizational strategy. Third, the strengths of mediational paths between "academic enjoyment →self-concept →English achievement" and "academic enjoyment → organizational strategy → English achievement" were also compared. The unstandardized estimate for the strength comparison of these two mediational paths was 20.163 and zero didn't occur in the BCa 95% CIs ([15.424, 25.869] and [15.457, 25.914]), evidencing that self-concept has a significantly larger mediation effect than organizational strategy. In terms of strength of the mediation effect, the mediation effects of self-concept, behavioral engagement, and organizational strategy decrease successively.

In the mediation model of the present study, the direct effect between academic enjoyment and English achievement is significant (BCa 95% CI were [-0.461, -0.049] and [-0.460, -0.049]) because the intervals didn't contain zero, indicating that behavioral engagement, self-concept, and organizational strategy play a partial mediating role in the relationship between these two constructs. Moreover, the mediation proportion, the total effect of academic enjoyment on English achievement via the three mediators, was 72.8% ($P_M = 0.644/(0.644+0.241) = 0.728$) (see Table 2), signifying that the mediating effect accounted for 72.8% of the total absolute

**Table 2. Bootstrap-computed analyses of the mediation effects of behavioral engagement, self-concept, and organizational strategy in the relationship between academic enjoyment and English achievement.**

| Types of paths | Estimate | Parameter | | Bootstrap 5000 times 95% CI | | | |
| --- | --- | --- | --- | --- | --- | --- | --- |
| | | | | bias-corrected | | percentile | |
| | | S.E. | Est./S.E. | Lower | Upper | Lower | Upper |
| 1 Enjoyment→Behavioral engagement→English achievement | 0.214 | 0.072 | 2.979 | 0.078 | 0.362 | 2.469 | 12.306 |
| 2 Enjoyment→Self-concept→English achievement | 0.525 | 0.059 | 8.937 | 0.423 | 0.657 | 0.657 | 21.858 |
| 3 Enjoyment→Organizational strategy→English achievement | -0.094 | 0.038 | -2.490 | -0.173 | -0.023 | -5.757 | -0.688 |
| Total indirect effects | 0.644 | 0.095 | 6.766 | 0.475 | 0.854 | 14.973 | 28.351 |
| Direct effects | -0.241 | 0.107 | -2.261 | -0.461 | -0.049 | -15.379 | -1.498 |
| Indirect effects comparison | | | | | | | |
| 1 Behavioral engagement–2 Self-concept | -10.137 | 3.046 | -3.328 | -16.096 | -4.248 | -16.103 | -4.252 |
| 1 Behavioral engagement–3 Organizational strategy | 10.026 | 3.035 | 3.303 | 4.600 | 16.592 | 4.587 | 16.587 |
| 2 Self-concept–3 Organizational strategy | 20.163 | 2.670 | 7.553 | 15.424 | 25.869 | 15.457 | 25.914 |

*Note*. The comparison of indirect effects was unstandardized output, while the remaining results were standardized output.

effect. The ratio-of-mediators ($R_M$) was 2.67 ($R_M$ = 0.644/0.241 = 2.67), which indicated that the total mediating effect of the three mediators was 2.67 times of the direct effect. Furthermore, the proportions of the three mediators' mediating effect to the total absolute effect were 19.9% ($R_{behavioral\ engagement}$ = 0.214/(0.214+0.525+0.094+0.241)), 48.9% ($R_{self\text{-}concept}$ = 0.525/(0.214+0.525+0.094+0.241)), 8.8% ($R_{organizational\ strategy}$ = 0.094/(0.214+0.525+0.094+0.241)) respectively.

## Structural invariance across genders

Multi-group SEM has been conducted to examine the measurement invariance of the between-construct relationships across genders. The measurement invariance across different groups (e.g., gender) is the logical prerequisite to testing the path coefficients invariance across the comparison groups. Therefore, establishing between-constructs measurement invariance across comparison groups would be first tested, and then examined whether the path coefficient has cross-group invariance.

Results of testing measurement invariance showed that the overall model fits were good, and ΔCFIs were 0.005 or smaller between two nested models (see Table 3), which were smaller than 0.01, indicating that configural, metric and scalar invariances were established in the proposed model for genders. Therefore, structural invariance was examined next.

We then examined the null hypothesis of equality of path coefficients in the proposed model across genders with the 'Model test' command in Mplus 8.3 [58]. If the Wald chi-square test was significant, we constrained one specific structural path to be equal at a time to identify paths with significant gender differences. If no significance was identified, we stopped the analysis. The differences in path estimates of the SEM model across genders were assessed by equally constraining the corresponding path coefficients for each group. The results of omnibus Wald tests across genders (in which gender as a control variable was excluded; Wald $\chi^2(7)$ = 7.427 and $p$ = 0.39) showed no significant difference in the path coefficients, indicating that there was no significant difference that all paths were equal for male and female students.

## Discussion

The present study aimed to investigate the association and the mediation mechanism between positive achievement emotion (i.e., enjoyment) and academic achievement in an EFL setting. With the rise of positive psychology, a growing number of studieshave concentrated on how positive achievement emotions relate to the subsequent academic results. Relations and mediation mechanisms between these two constructs have been comparatively less studied in the EFL context, especially those that took secondary school students as participants. EFL-related Data of a sample of Chinese secondary school students in seventh and eighth grades were collected. The latent SEM was conducted to calculate the correlation coefficients and model relationships among the variables for data analysis.

**Table 3. Fit indices for measurement invariance tests of the model across genders.**

| Model | $\chi^2$ | df | CFI | ΔCFI | TLI | RMSEA | SRMR |
|---|---|---|---|---|---|---|---|
| M2[a]: Configural model | 532.259 | 292 | 0.960 | - | 0.954 | 0.056 | 0.054 |
| M3[a]: Metric invariance | 552.242 | 307 | 0.960 | 0.000 | 0.955 | 0.055 | 0.060 |
| M4[a]: Scalar invariance | 596.001 | 326 | 0.955 | 0.005 | 0.953 | 0.056 | 0.061 |

*Note*: CFI = comparative fit index (>0.90); TLI = Tucker–Lewis index (>0.90); RMSEA = root mean square error of approximation (<0.10); SRMR = standardized root mean square residual (<0.08) [63].

[a] Fit index for measurement invariance tests of the model across genders ($N_{boy}$ = 280; $N_{girl}$ = 248).

The hypothesis that academic enjoyment has a significant positive effect on EFL achievement ($H_1$) was supported. A large body of research has been inspired by CVT, showing that academic enjoyment is positively correlated with academic achievement [e.g., 5, 11, 64, 65]. The present study contributes to the literature by offering empirical evidence that positive achievement emotions such as enjoyment have a predictive effect on academic achievement based on samples of Chinese secondary school students' EFL learning. However, academic enjoyment positively influences English achievement via the mediators of behavioral engagement, academic self-concept, and organizational strategy. However, academic enjoyment alone is not enough to guarantee English achievement because behavioral engagement, self-concept, and organizational strategy are also necessary [66]. This shows that the mechanism of achievement emotions on academic achievement is complex, and more research is needed to deepen the understanding of how achievement emotions affect students' academic engagement.

The hypotheses that behavioral engagement ($H_2$), self-concept ($H_3$), and organizational strategy ($H_4$) are the mediators between academic enjoyment and English achievement were also confirmed. Researchers have explored the mediation mechanism between enjoyment and achievement across a wide range of subject domains [e.g., 4, 10, 31, 67, 68]. And a succession of mediators such as achievement goals, academic engagement, academic self-concept, psychological capital, and learning strategy was certified [69–72]. However, most of the existing studies only discussed one single mediator between these two constructs [73], and few studies have simultaneously investigated multiple mediators. Furthermore, this study also calculated and compared the path coefficients of the three mediators. In this study, $P_M$ (the ratio of the indirect effect to the total effect) $_=$ .728 represents the total mediation effect size of enjoyment on English achievement was 72.8%. Self-concept took up the highest mediation effect, accounting for 48.9% of the total mediation effect. While the mediation effect sizes of behavioral engagement and organizational strategy decreased sequentially, accounting for 19.9% and 8.8%, respectively. The finding that academic enjoyment predicted English achievement via the mediators of self-concept, behavioral engagement, and organizational strategy supports the existing studies documenting this linkage in Chinese secondary school students' EFL learning [e.g., 2, 5, 10, 11]. It is a novel contribution to the literature that the mediating effect of variables (i.e., self-concept, behavioral engagement, and organizational strategy) between academic enjoyment and English achievement differed in samples of Chinese secondary school students' EFL learning. Accordingly, the findings suggest that in addition to evoking students' organizational strategy, their EFL-related self-concept and behavioral engagement are more requested to give full play to the predictive effect of academic enjoyment on English achievement.

Both measurement invariance and path coefficients invariance of the assumed multi-mediation model across genders were verified. The multi-group confirmatory factor analysis and multi-group SEM analysis showed that both factor loadings and item intercepts were equivalent across genders, indicating that solid measurement invariance was established across genders. In addition, the omnibus Wald test across genders (Wald $\chi^2(7) = 7.427$ and $p = 0.39$) indicated that all paths were equally applicable to both male and female secondary school students.

## Implications, limitations, and directions for future research

This study has both theoretical and practical implications for the relevance of student-perceived enjoyment to their academic achievement in an EFL setting. CVT posits that achievement emotions (e.g., enjoyment) are generated by the synergistic effects of control and value appraisals [11]. Moreover, control and value are considered to be two malleable constructs

[9, 73], which implies that important socializers such as educational practitioners and parents could exert a positive influence on children's control and value appraisals and thereby enhance their positive emotions (e.g., enjoyment). Given the positive correlation between perceived enjoyment and academic achievement in the EFL context, we would anticipate that increasing students' perceived enjoyment in English learning would improve their English achievement. Specifically, educational practitioners were suggested to adopt an open, extrovert, and agreeable attitude in EFL teaching [74], and parents were advised to praise their children [75, 76], these measures would be beneficial in enhancing students' academic enjoyment and thus improve their EFL achievement. In terms of theory, this study contributes to the achievement emotion literature by providing empirical evidence that positive achievement emotions such as enjoyment could affect academic achievement directly or indirectly via the mediators of behavioral engagement, self-concept, and organizational strategy.

Although we provided empirical evidence supporting academic enjoyment as a direct and indirect predictor of academic achievement among secondary school students in an EFL setting, three limitations need to be addressed. First, we solely focused on the correlation between enjoyment and academic achievement, as being the most common but under-researched positive emotion [8, 64]. However, other positive emotions such as hope, pride, and relief yet need to be discussed in the EFL setting. Future research on positive achievement emotions and academic achievement should consider these emotions. Second, the correlational nature of this study determines that a causal relationship between the variables of enjoyment and academic achievement cannot be drawn. A longitudinal design would be preferable for future research to establish causality between enjoyment and academic achievement. Third, while this study explored the mediation effects of behavioral engagement, self-concept, and organizational strategy in the relationship between academic enjoyment and English achievement, it did not exhaust all the mediators between these two constructs. For example, coping styles, achievement goals, psychological flexibility, and psychological capital [67, 70, 71, 77, 78] are also possible mediators that need to be verified in future research.

## Conclusion

This study aimed to explore the association between academic enjoyment and English achievement and the mediation mechanism between these two constructs in a sample of students aged 12 to 14 years. The findings showed that students' English-related enjoyment had direct and indirect predictive effects on their English achievement. In addition, we also found that self-concept, behavioral engagement, and organizational strategy play a mediating role between perceived enjoyment and English achievement, and the mediation effects decrease successively. This study significantly enriched the extant literature on academic enjoyment in the EFL domain. Theoretically, we provided empirical evidence for the hypothesis that positive emotions (i.e., enjoyment) promote academic achievement in EFL settings. The mediation mechanism between academic enjoyment and English achievement was also explored, which figured out that behavioral engagement, self-concept, and organizational strategy are the effective paths for enjoyment to affect academic achievement in the EFL context.

## Acknowledgments

We thank the principal, teachers, and students for their support and participation.

## Author Contributions

**Conceptualization:** Xia Kang, Yajun Wu.

**Formal analysis:** Xia Kang.

**Investigation:** Yajun Wu.

**Methodology:** Xia Kang.

**Validation:** Xia Kang.

**Writing – original draft:** Yajun Wu.

**Writing – review & editing:** Xia Kang.

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
