## [Decision Letter · Decision Letter 0]

2 Mar 2022

PONE-D-21-30639

Academic Enjoyment, Behavioral Engagement, Self-concept, Organizational Strategy and Achievement in EFL Setting: A Multiple Mediation Analysis

PLOS ONE

Dear Dr. Wu,

Thank you for submitting your manuscript to PLOS ONE. After careful consideration, we feel that it has merit but does not fully meet PLOS ONE’s publication criteria as it currently stands. Therefore, we invite you to submit a revised version of the manuscript that addresses the points raised during the review process.

We look forward to receiving your revised manuscript.

Kind regards,

Trinidad Garcia, PhD

Academic Editor

PLOS ONE

https://journals.plos.org/plosone/s/fileid=ba62/PLOSOne_formatting_sample_title_authors_affiliations.pdf".

Reviewers' comments:

Reviewer's Responses to Questions

**Comments to the Author**

1. Is the manuscript technically sound, and do the data support the conclusions?

Reviewer #1: Yes

Reviewer #2: Yes

Reviewer #3: Yes

Reviewer #4: Yes

2. Has the statistical analysis been performed appropriately and rigorously? 

Reviewer #1: Yes

Reviewer #2: Yes

Reviewer #3: I Don't Know

Reviewer #4: Yes

3. Have the authors made all data underlying the findings in their manuscript fully available?

Reviewer #1: Yes

Reviewer #2: No

Reviewer #3: Yes

Reviewer #4: Yes

4. Is the manuscript presented in an intelligible fashion and written in standard English?

Reviewer #1: Yes

Reviewer #2: Yes

Reviewer #3: Yes

Reviewer #4: Yes

5. Review Comments to the Author

Reviewer #1: Since it is rare to find the studies examning why acaemic enjoyment can predict positive academic achievement, this study focused on investigating whether behavioral engagement, self-concept, and organizational strategy mediated relations between academic enjoyment and achievement in the English as Foreign Language (EFL) setting, in a sample of secondary school students.

The background was presented reasonably. It is also found that the literature review covered main points of the study. As for the methods, the detailed information was damonstrated clearly. The way for presenting the results is consistent with the purposes of the study. Besides, the fact and argument were integrated and written systematically.

Therefore, this manuscript was accepted to publish.

Reviewer #2: Academic Enjoyment, Behavioral Engagement, Self-concept, Organizational Strategy and Achievement in EFL Setting: A Multiple Mediation Analysis

This study aimed to investigate whether behavioural engagement, self-concept, and organizational strategy mediated relations between academic enjoyment and achievement in an English as a foreign language setting in China. Overall, 528 learners participated in the study. The paper has several strengths (e.g., a decent sample size and strong data analysis section). However, I believe there are a number of issues that need to be addressed before considering this paper for publication. Please see my comments below which are in no particular order of importance.

1.Abstract:

•The abstract needs to be improved.

a.The first two sentences need to be revised - the authors need to highlight the need for studies exploring the mechanisms through which enjoyment predicts academic achievement.

b.The abstract does not include all the information needed for the reader. E.g., there is no indication where this study took place and age/gender of the participants.

c.In the abstract, the authors just report the percentages of the mediating effects. I would suggest that they simplify this section and just directly state whether the proposed mediators mediated the relations between enjoyment and achievement.

2.Introduction and literature review

•In the introduction, the authors make false claims – e.g.,

“However, except for a few studies focusing on language anxiety, positive achievement emotions, for example, academic enjoyment, was seldom explored in the EFL setting”

a.Anxiety is not a positive emotion.

b.There are at least 10 studies conducted by Prof Jean-Marc Dewaele and many other recent publications by other scholars. In fact, language enjoyment is the most widely studied positive emotion in the field.

•I expected to read more information about why language enjoyment or positive emotions more generally matter for language achievement. There are loads of studies on emotions in the field – so rather than claiming that there are little research, I would suggest that the authors highlight the importance of the topic and why we need to explore the mechanisms through which enjoyment predicts language achievement.

•It is not clear anywhere in the manuscript how the authors decided on the mediators. Why did they choose “Behavioural Engagement, Self-concept, Organizational Strategy” as the mediators? Why was it important to explore these three constructs together?

•Did the authors use the CVT as their theoretical framework? Then, what does the CVT say about emotions predicting the achievement? They need to expand on this a bit more.

•The relations between the study variables should be better articulated. The authors say, for example,: “Goetz et al. (2008) examined the mediating effect of academic self-concept on mathematics-related achievement and enjoyment.” In this study, enjoyment seems to be the outcome and self-concept is the predictor – how do the authors explain enjoyment being predictor and self-concept the outcome/mediator in their study?

•There needs to be a section or at least mention of a few studies showing the relations between the mediators.

3.Method & Results

•In 2.3., the authors repeat that “However, studies have yet to investigate how positive achievement emotions (e.g., enjoyment) promote academic achievement in the EFL context”. Again, there are studies investigating these relations already. I suggest they highlight the strengths of their study and why it is needed/different/important.

•The section 3.1. needs to be revised. Some sections are not clear – e.g., students “…were told that they could choose not to fill out the questionnaire without affecting them”.

•Also, in Section 3.1., the authors need to explain the context a bit more. Currently, the only information we have about the participants is that they are secondary school students based in China. I think we need more information about their context to make better sense of why enjoyment matters for this particular group of learners.

•Measurement invariance should be reported before the mediation analyses are presented.

•Regarding the measurement invariance, I wonder why the authors did not test residual variance. They need to explain it.

•It is difficult to follow 4.3.2. I suggest the authors do not include the CIs in text as they are already presented in the table.

4.Discussion and Conclusions

•Discussion needs to be improved.

E.g. the authors state that: “With the rise of positive psychology, a growing number of studies, especially in STEM domains, have concentrated on how positive achievement emotions relate to the subsequent academic results.” This also applies to the field of SLA.

•The authors claim that “However, most of the existing studies only discussed one single mediator between these two constructs, and few studies have simultaneously investigated multiple mediators.” I think they need to provide the relevant citations here. Which studies do they refer to?

•The authors need to improve the implications section. Currently, the only suggestion they give is to increase enjoyment in the EFL classrooms. I think there are more implications to discuss here.

5.Other issues:

-Proofreading is needed.

-Citations and references need to be checked.

Reviewer #3: I appreciate the effort made in this paper. I believe that this study represents a crucial topic in the EFL context because the number of studies that examine the psychological state of students and its relation to their achievement in the EFL setting is limited. The present study reveals that academic enjoyment and achievement are strongly related. To study this relation, three mediating factors are used, which are self-concept, behavioral engagement and organizational strategy. The objectives of the study are clearly stated and supported by good evidence and examples. Also, the use of the mediating mechanism is a good strategy to show how achievement emotions and academic achievement are related. Moreover, the study is well organized and consistent. Finally, the researchers mention a number of limitations and suggestions for future research at the end of the study.

Note: I must say that I don’t have the adequate knowledge to revise the provided statistics and tables. However, I suggest more explanation on the provided results. The study mentions that a number of items from different questionnaires is used for each variable (without stating them in the study and merely giving one example on each). For example, researchers indicate that "academic self-concept was measured with 5 items", but only mention one example “I am good at English”. Also, the results are immediately drawn based on the overall percentages of the major category of each factor. The provided discussion is based on a mere statistical analysis lacking interpretive analysis on the given statistics.

Reviewer #4: The study presents the results of novel and original research.

The language is very good and is written in standard English.

The methodology is acceptable: Experiments, statistics, and other analyses are performed to a high technical standard and are described in sufficient detail.

The authors presented their results, discussion and conclusions in an appropriate manner.

Published as-is/

6. PLOS authors have the option to publish the peer review history of their article (what does this mean?). If published, this will include your full peer review and any attached files.

Reviewer #1: **Yes: **Dr. Urarat Parnrod

Reviewer #2: No

Reviewer #3: **Yes: **Ruba Murad Mahfouz Siaj

Reviewer #4: **Yes: **Mohammed Farrah

---

## [Author Response · Author response to Decision Letter 0]

4 Apr 2022

Academic Enjoyment, Behavioral Engagement, Self-concept, Organizational Strategy and Achievement in EFL Setting: A Multiple Mediation Analysis

This study aimed to investigate whether behavioural engagement, self-concept, and organizational strategy mediated relations between academic enjoyment and achievement in an English as a foreign language setting in China. Overall, 528 learners participated in the study. The paper has several strengths (e.g., a decent sample size and strong data analysis section). However, I believe there are a number of issues that need to be addressed before considering this paper for publication. Please see my comments below which are in no particular order of importance.

1. Abstract:

• The abstract needs to be improved.

a. The first two sentences need to be revised - the authors need to highlight the need for studies exploring the mechanisms through which enjoyment predicts academic achievement. 

b. The abstract does not include all the information needed for the reader. E.g., there is no indication where this study took place and age/gender of the participants. 

c. In the abstract, the authors just report the percentages of the mediating effects. I would suggest that they simplify this section and just directly state whether the proposed mediators mediated the relations between enjoyment and achievement. 

Authors’ response: Thank you very much for your valuable comments and suggestions. First, to highlight the need for exploring the mechanism between academic enjoyment and academic achievement, research background and research gaps were added in the revised edition. Second, according to your recommendation, the gender and nationality of the participants were introduced in the revised edition. Third, we accept your suggestion and simplify the finding part of the abstract. 

2. Introduction and literature review 

• In the introduction, the authors make false claims – e.g.,

“However, except for a few studies focusing on language anxiety, positive achievement emotions, for example, academic enjoyment, was seldom explored in the EFL setting”

a. Anxiety is not a positive emotion.

b. There are at least 10 studies conducted by Prof Jean-Marc Dewaele and many other recent publications by other scholars. In fact, language enjoyment is the most widely studied positive emotion in the field. 

• I expected to read more information about why language enjoyment or positive emotions more generally matter for language achievement. There are loads of studies on emotions in the field – so rather than claiming that there are little research, I would suggest that the authors highlight the importance of the topic and why we need to explore the mechanisms through which enjoyment predicts language achievement. 

• It is not clear anywhere in the manuscript how the authors decided on the mediators. Why did they choose “Behavioural Engagement, Self-concept, Organizational Strategy” as the mediators? Why was it important to explore these three constructs together? 

• Did the authors use the CVT as their theoretical framework? Then, what does the CVT say about emotions predicting the achievement? They need to expand on this a bit more. 

• The relations between the study variables should be better articulated. The authors say, for example,: “Goetz et al. (2008) examined the mediating effect of academic self-concept on mathematics-related achievement and enjoyment.” In this study, enjoyment seems to be the outcome and self-concept is the predictor – how do the authors explain enjoyment being predictor and self-concept the outcome/mediator in their study? 

• There needs to be a section or at least mention of a few studies showing the relations between the mediators. 

Authors’ response: Thank you very much for your comments and suggestions. 

1. Recently, scholars such as Prof. Jean-Marc Dewaele and Dr. Li studied the effect of achievement emotions (e.g., enjoyment, anxiety, and boredom) on academic performance in EFL context. One the one hand, their research has beneficial inspiration for the present study. On the other hand, the claim “However, except for a few studies focusing on language anxiety, positive achievement emotions, for example, academic enjoyment, was seldom explored in the EFL setting”. Besides, anxiety is one kind of negative achievement emotions (i.e., anxiety, hopelessness, boredom, shame, and anger). Based on reviewers’ comments, we rewrote this sentence. 

2. The present study was conducted for three reasons. First, previous studies indicated that academic enjoyment has a moderate to strong positive relationship with academic achievement (e.g., Goetz et al., 2008) and the change in the level of academic enjoyment would lead to the corresponding change in academic performance (Ahmed et al., 2010). Second, inspired by the positive psychology movement in EFL context (see Li, 2020), exploring the mediating effects of academic enjoyment on academic achievement would help educators to clarify the indirect effect of academic enjoyment on academic achievement. Specifically, self-concept accounted for 48.9% of the total effect, while behavioral engagement and organizational strategy accounted for 19.8% and 8.8%, respectively. Third, mediation effect analysis would provide information on whether the relationship between independent variable (i.e., academic enjoyment) and dependent variable (i.e., academic performance) is partly or wholly attributable to mediators (i.e., academic self-concept, organizational strategy, and behavioral engagement) (Mackinnon, 2008), which would be helpful for educators to understand the relationship between independent variable(s) and dependent variable(s).

3. The present study focused on the mediating effects of behavioral engagement, organizational strategy and academic self-concept between academic enjoyment and academic performance in the EFL context. There are two reasons for the selection of mediators between academic enjoyment and academic performance. One is that previous studies have confirmed the mediating effects of these three variables between achievement emotions and academic performance in other disciplines (e.g., mathematics)(Fredricks et al., 2004; Goetz et al., 2008; Pinxten et al., 2014). In view of the domain-specific nature of achievement emotions (e.g., Pekrun et al., 2006), it is meaningful to verify the mediating effects of these three variables between academic enjoyment and academic achievement in the EFL context. The other one is that self-concept, organizational strategy, and behavioral engagement are significant indicators of academic achievement (e.g., Froiland & Oros, 2014; Habók & Magyar, 2018), which would reveal the mechanism by which academic enjoyment affect academic performance. 

4. Thank you for your recommendation. The present study was based on the control-value theory (CVT) that proposed by Pekrun et al. (2002) and Pekrun et al. (2006). We accept your suggestion and add the effect of achievement emotions on academic performance by the CVT (see Section 2.1). 

5. Both theoretical framework and empirical evidence demonstrated the reciprocal relationships between achievement emotions and academic self-control (e.g., Clem et al., 2021; Pekrun et al., 2007; Pinxten et al., 2014). According to your suggestions, we added literature on the reciprocal relationships between achievement emotions and academic self-control (see Section 2.2.2). 

3. Method & Results 

• In 2.3., the authors repeat that “However, studies have yet to investigate how positive achievement emotions (e.g., enjoyment) promote academic achievement in the EFL context”. Again, there are studies investigating these relations already. I suggest they highlight the strengths of their study and why it is needed/different/important. 

• The section 3.1. needs to be revised. Some sections are not clear – e.g., students “…were told that they could choose not to fill out the questionnaire without affecting them”. 

• Also, in Section 3.1., the authors need to explain the context a bit more. Currently, the only information we have about the participants is that they are secondary school students based in China. I think we need more information about their context to make better sense of why enjoyment matters for this particular group of learners.

• Measurement invariance should be reported before the mediation analyses are presented.

• Regarding the measurement invariance, I wonder why the authors did not test residual variance. They need to explain it. 

• It is difficult to follow 4.3.2. I suggest the authors do not include the CIs in text as they are already presented in the table. 

Authors’ response: We appreciate your high-value comments and suggestions. 

1. In recent years, researchers such as Prof. Dewaele and Dr. Li conducted studies on the relationships between positive achievement emotions and academic performance in field of English as a foreign language (e.g., Dewaele et al., 2018; Li, 2020; Li et al., 2020). These studies provide valuable insights to the present study. However, the claim that “However, studies have yet to investigate how positive achievement emotions (e.g., enjoyment) promote academic achievement in the EFL context” is a bit strong. We accept reviewer’s suggestions and revised this statement. The revised sentence would emphasis the importance of the present study rather than the lack of related research.

2. We revised the sentence. E.g., “…were told that they could choose not to fill out the questionnaire without affecting them”. Besides, more information related to the participants was also provided in Section 3.1. 

3. The major aim of the present study was to test the mediating effect of behavioral engagement, organizational strategy, and academic self-concept between academic enjoyment and academic performance in a sample of 528 Chinese secondary school students. Ensuingly, Multi-group SEM has been carried out to exam the measurement invariance of the between-constructs relationships across genders. That is, boy students and girl students were treated as a whole in the mediation analyses. Wang and Wang (2019: 255) listed four criteria for measurement invariance: (1) pattern invariance; (2) weak measurement invariance; (3) strong measurement invariance (i.e., metric and scalar invariance); (4) strict measurement invariance (i.e., metric, scalar, and error variance invariance). The present study adopted the strong measurement invariance criterion instead of the strict measurement invariance criterion. There are two reasons for this choice. Measurement invariance in most of the previous studies include configural, metric, and scalar steps and only a few study test the residual invariance (Putnick & Bornstein, 2016), which means that existing studies mainly adopted the strong measurement invariance criterion. Besides, Bentler (2006) documented that residual invariance is unnecessary for many disciplines. Although a few studies tested the residual invariance for the purpose of examing the item reliabilities across groups (Schmitt et al., 1984), however, “this is true only if factor variances are invariant across groups” (Wang & Wang, 2019: 256).

4. Discussion and Conclusions 

• Discussion needs to be improved. 

E.g. the authors state that: “With the rise of positive psychology, a growing number of studies, especially in STEM domains, have concentrated on how positive achievement emotions relate to the subsequent academic results.” This also applies to the field of SLA. 

• The authors claim that “However, most of the existing studies only discussed one single mediator between these two constructs, and few studies have simultaneously investigated multiple mediators.” I think they need to provide the relevant citations here. Which studies do they refer to? 

• The authors need to improve the implications section. Currently, the only suggestion they give is to increase enjoyment in the EFL classrooms. I think there are more implications to discuss here. 

Authors’ response: Thank you for your valuable suggestions. According to your suggestions, implication section as well as some statements in Discussion section were rewrote.

5. Other issues: 

- Proofreading is needed. 

- Citations and references need to be checked. 

Authors’ response: Thank you very much for your suggestions. Authors proofread the whole manuscript and the references were checked one by one.

References

Ahmed, W., Minnaert, A., van der Werf, G., & Kuyper, H. (2010). Perceived social support and early adolescents’ achievement: The mediational roles of motivational beliefs and emotions. Journal of Youth and Adolescence, 39(1), 36–46. https://doi.org/10.1007/s10964-008-9367-7

Bentler, P. M. (2006). EQS 6 structural equations program manual. Multivariate Software.

Clem, A. L., Hirvonen, R., Aunola, K., & Kiuru, N. (2021). Reciprocal relations between adolescents’ self-concepts of ability and achievement emotions in mathematics and literacy. Contemporary Educational Psychology, 65, 101964. https://doi.org/10.1016/j.cedpsych.2021.101964

Dewaele, J.-M., Witney, J., Saito, K., & Dewaele, L. (2018). Foreign language enjoyment and anxiety: The effect of teacher and learner variables. Language Teaching Research, 22(6), 676–697.

Fredricks, J. A., Blumenfeld, P. C., & Paris, A. H. (2004). School engagement: Potential of the concept, state of the evidence. Review of Educational Research, 74(1), 59–109. https://doi.org/10.3102/00346543074001059

Froiland, J. M., & Oros, E. (2014). Intrinsic motivation, perceived competence and classroom engagement as longitudinal predictors of adolescent reading achievement. Educational Psychology, 34(2), 119–132. https://doi.org/10.1080/01443410.2013.822964

Goetz, T., Frenzel, A. C., Hall, N. C., & Pekrun, R. (2008). Antecedents of academic emotions: Testing the internal/external frame of reference model for academic enjoyment. Contemporary Educational Psychology, 33(1), 9–33. https://doi.org/10.1016/j.cedpsych.2006.12.002

Habók, A., & Magyar, A. (2018). The effect of language learning strategies on proficiency, attitudes and school achievement. Frontiers in Psychology, 8(JAN), 1–8. https://doi.org/10.3389/fpsyg.2017.02358

Li, C. (2020). A positive psychology perspective on Chinese EFL students’trait emotional intelligence, foreign language enjoyment and EFL learning achievement. Journal of Multilingual and Multicultural Development, 41(3), 246–263. https://doi.org/10.1080/01434632.2019.1614187

Li, C., Dewaele, J. M., & Jiang, G. (2020). The complex relationship between classroom emotions and EFL achievement in China. Applied Linguistics Review, 11(3), 485–510. https://doi.org/10.1515/applirev-2018-0043

Mackinnon, D. P. (2008). Introduction to statistical mediation analysis. Erlbaum. https://doi.org/10.4324/9780203809556

Pekrun, R., Elliot, A. J., & Maier, M. A. (2006). Achievement goals and discrete achievement emotions: A theoretical model and prospective test. Journal of Educational Psychology, 98(3), 583–597. https://doi.org/10.1037/0022-0663.98.3.583

Pekrun, R., Frenzel, A. C., Goetz, T., & Perry, R. P. (2007). The control-value theory of achievement emotions: An integrative approach to emotions in education. In P. A. Schutz & R. Pekrun (Eds.), Emotion in education (pp. 13–36). Academic Press.

Pekrun, R., Goetz, T., Titz, W., & Perry, R. P. (2002). Academic emotions in students’ self-regulated learning and achievement: A program of qualitative and quantitative research. Educational Psychologist, 37(2), 91–105. https://doi.org/10.1207/S15326985EP3702_4

Pinxten, M., Marsh, H. W., De Fraine, B., Van Den Noortgate, W., & Van Damme, J. (2014). Enjoying mathematics or feeling competent in mathematics? Reciprocal effects on mathematics achievement and perceived math effort expenditure. British Journal of Educational Psychology, 84, 152–174. https://doi.org/10.1111/bjep.12028

Putnick, D. L., & Bornstein, M. H. (2016). Measurement invariance conventions and reporting: The state of the art and future directions for psychological research. Developmental Review, 41, 71–90. https://doi.org/10.1016/j.dr.2016.06.004

Schmitt, N., Pulakos, E. D., & Lieblein, A. (1984). Comparison of three techniques to assess group-level beta and gamma change. Applied Psychological Measurement, 8(3), 249–260. https://doi.org/10.1177/014662168400800301

Wang, J., & Wang, X. (2019). Structural equation modeling: Applications using Mplus. Wiley.

---

## [Editor Report · Decision Letter 1]

8 Apr 2022

Academic Enjoyment, Behavioral Engagement, Self-concept, Organizational Strategy and Achievement in EFL Setting: A Multiple Mediation Analysis

PONE-D-21-30639R1

Dear Dr. Wu,

We’re pleased to inform you that your manuscript has been judged scientifically suitable for publication and will be formally accepted for publication once it meets all outstanding technical requirements.

Kind regards,

Trinidad Garcia, PhD

Academic Editor

PLOS ONE
---

## [Editor Report · Acceptance letter]

13 Apr 2022

PONE-D-21-30639R1 

*Academic enjoyment, behavioral engagement, self-concept, organizational strategy and achievement in EFL setting: A multiple mediation analysis*

Dear Dr. Wu:

I'm pleased to inform you that your manuscript has been deemed suitable for publication in PLOS ONE. Congratulations! Your manuscript is now with our production department. 

Kind regards, 

on behalf of

Dr. Trinidad Garcia 

Academic Editor

PLOS ONE